# SPARSE FEATURE ROUTING FOR TABULAR LEARNING

## ABSTRACT

The landscape of high-performance tabular learning is often framed as a choice between the efficiency of gradient-boosted trees and the performance of deep architectures, which increasingly rely on heavy, monolithic backbones to model feature interactions. We argue that this monolithic design overlooks a critical inductive bias: the inherent sparsity and modularity of tabular data. To address this, we introduce the Sparse Feature Routing Network (SFR Net), an architecture that decomposes computation into independent feature experts controlled by an entropy-regularized router, coupled with a low-rank module to capture non-additive dependencies. We evaluate SFR Net across 14 heterogeneous benchmarks, including standard datasets, high-dimensional multiclass tasks, and regression problems. Empirically, SFR Net demonstrates predictive performance competitive with, and often superior to, state-of-the-art deep tabular models and gradient-boosted ensembles. Beyond raw performance, SFR Net offers three distinct structural advantages: (1) efficiency, requiring up to $24\times$ fewer parameters and training $30\times$ faster than tabular Transformers; (2) intrinsic sparsity, dynamically activating only a small fraction of features per instance; and (3) faithful interpretability, where deletion tests confirm that the learned routing weights serve as reliable, causal instance-level attributions. These results position sparse feature routing as a lightweight, transparent, and high-performance alternative to dense tabular foundation models.

## 1 INTRODUCTION

Tabular data are among the most widespread data modalities, arising in domains such as healthcare, finance, and the social sciences. Despite their ubiquity, learning effective representations from tabular datasets remains a fundamental challenge. Unlike image, text, or audio inputs, tabular records consist of heterogeneous features of varying types and scales, often with weak or irregular dependencies. These characteristics limit the transfer of inductive biases that have powered deep learning breakthroughs in unstructured modalities. As a result, classical approaches such as gradient-boosted decision trees remain dominant in practice, while neural networks have historically struggled to consistently outperform them.

Recent research has begun to narrow this gap by improving training strategies, designing novel architectures, and exploring self-supervised objectives tailored to tabular domains. Advances in normalization, regularization, and feature encoding have enhanced the robustness of deep models, while architectural innovations such as attention mechanisms and modular processing blocks have sought to capture complex feature interactions. In parallel, representation learning methods have shown that self-supervision can extract informative embeddings without labels, improving downstream classification and regression tasks. Yet, despite this progress, a core difficulty persists: most existing models still treat tabular inputs monolithically, processing all features through shared encoders. This design blurs the role of individual columns, complicates interpretability, and can reduce robustness in the presence of irrelevant or noisy features.

In this work, we propose the **Sparse Feature Routing Network (SFR Net)**, a new architecture designed to address these limitations. Our approach decomposes each feature into a specialized expert network, ensuring that heterogeneous columns are modeled according to their individual distributions. These experts are then dynamically composed by a lightweight router that assigns sparse, entropy-regularized attention weights, selecting only the most informative features for each instance. This design provides three key advantages: (i) principled handling of heterogeneity through feature-

wise specialization, (ii) efficiency and robustness via sparse routing that scales with feature count and resists irrelevant inputs, and (iii) native interpretability, as the router's weights yield direct, transparent per-instance attributions without the need for post-hoc analysis.

Beyond raw performance, the novelty of SFR Net lies in introducing sparse, feature-level routing as an inductive principle for tabular learning. While previous approaches have relied on either dense transformations of all features or indirect mechanisms such as mask prediction, our framework directly encodes the idea that each column should contribute selectively and independently to the decision process. This perspective not only improves predictive accuracy but also aligns with how practitioners naturally interpret tabular data—by analyzing the marginal and conditional relevance of individual features. In doing so, SFR Net provides a step toward architectures that are not only competitive with established baselines but also inherently interpretable and resilient under distributional shifts.

Our contributions can be summarized as follows:

- **Feature-wise specialization with sparse routing.** We introduce a feature-expert decomposition with an entropy-regularized router that performs instance-conditioned, sparse selection, providing a principled inductive bias for heterogeneous tabular data.
- **Competitive accuracy without pre-training.** SFR Net achieves performance competitive with or superior to recent deep learning methods across a range of tabular benchmarks, while avoiding complex self-supervised pre-training and maintaining a simple, efficient architecture.
- **Native interpretability and efficiency.** The router's sparse weights yield direct, explicit per-instance attributions from the model itself, removing the need for post-hoc methods. The architecture scales linearly with feature count, allowing effective training even on CPUs.

## 2 RELATED WORK

### 2.1 SPARSE FEATURE-EXPERT ROUTING

Conditional computation has long been studied as a way to scale neural models while improving efficiency and interpretability. The Mixture-of-Experts (MoE) paradigm (Jacobs et al., 1991; Shazeer et al., 2017; Lepikhin et al., 2021) routes inputs to one of several interchangeably-parameterized sub-networks, while developments in differentiable gating, such as `sparsemax` and `entmax` (Martins & Astudillo, 2016; Peters et al., 2019), provide tools to enforce controllable top-$k$ sparsity. Our work adopts these ideas but reframes the target of sparsity. Instead of routing between global experts that all see the entire input, we introduce *feature-expert routing*, where each input feature is assigned its own dedicated expert network and sparsity is applied directly at the feature level.

Low-rank interaction modules such as factorization machines (Rendle, 2010), cross networks (Wang et al., 2021), and high-order interaction blocks play a complementary role by capturing higher-order dependencies efficiently. In most prior work, however, these interactions are embedded in monolithic backbones, making it difficult to relate them back to individual features. SFR Net departs from this tradition by combining instance-wise sparse routing over feature-specific experts with a rank-controlled mixer, explicitly tying interaction capacity to a small number of low-rank factors.

### 2.2 TABULAR DEEP LEARNING AND FOUNDATIONAL MODELS

A large body of work has explored deep architectures for supervised learning on tabular data, including attention-based models (e.g., TabTransformer, FT-Transformer) (Huang et al., 2020; Gorishniy et al., 2021), retrieval-augmented models (Somepalli et al., 2021; Gorishniy et al., 2024; Ye et al., 2024), and capsule-like or compositional architectures (Chen et al., 2023). Generalized additive models and Neural Additive Models (NAMs) (Agarwal et al., 2021) emphasize per-feature structure and interpretability by enforcing a global additive decomposition, but do not support sample-specific expert selection or controlled non-additive interactions.

More recently, *tabular foundation models* have emerged, aiming for broad coverage across diverse datasets. TabPFN and its successors (Hollmann et al., 2023) use transformers trained in a meta-

learning fashion to amortize inference over synthetic tasks, while RealMLP, TabDPT, TabICL, and LimiX refine MLP-based designs with better regularization, pre-training objectives, or in-context learning strategies. TabM (Gorishniy et al., 2025) provides an especially strong and practical MLP-based baseline: by combining parameter-efficient BatchEnsemble-style ensembling with per-feature embeddings and careful tuning on a 46-dataset benchmark, TabM achieves the best average rank among deep tabular models and competes with GBDTs in both performance and efficiency. These results show that well-engineered MLPs are stronger than many attention- and retrieval-based architectures, and they frame simple MLPs and TabM as robust default baselines.

Our work is complementary to this line. Foundational tabular models typically operate on dense, shared feature representations and do not implement instance-wise sparse routing over feature-specific experts. SFR Net instead treats sparsity and feature-wise modularity as the central architectural principle. On OpenML-CC18, we compare against the state-of-the-art landscape and report that SFR Net is competitive with strong deep baselines.

## 2.3 Representation Learning for Tabular Data

Self-supervised learning (SSL) has been successfully adapted to tabular domains through objectives such as masked feature reconstruction, contrastive learning, and sub-sampling tasks (Yoon et al., 2020; Bahri et al., 2022; Ucar et al., 2021). These approaches typically pre-train a monolithic encoder and then fine-tune it for downstream supervised tasks, often using transformers or ResNet-like backbones. Recent work on tabular JEPA-style models further explores predictive objectives that operate on structured partitions of the input. While powerful, such methods often require substantial pre-training compute and do not expose feature-wise structure directly.

SFR Net instead relies on a single supervised training phase and encodes inductive biases directly in the architecture: feature-wise experts, sparse instance-wise routing, and a low-rank interaction head. Our experiments compare SFR Net against SSL-enhanced ResNet models on our core benchmarks and show that explicit feature-level structure can match or surpass SSL backbones on several datasets, despite the absence of pre-training.

**Relation to Neural Additive Models.** Neural Additive Models (NAMs) enforce a globally additive decomposition of the form $f(x) = \sum_j f_j(x_j)$, which yields strong interpretability at the cost of limited interaction capacity. SFR Net differs in two fundamental ways. First, its router introduces *instance-wise sparsity*: for each sample, only a subset of feature experts is activated, whereas NAMs do not support sample-specific expert selection. Second, the low-rank interaction head introduces controlled non-additive interactions among the routed features, breaking strict additivity while keeping interaction capacity bounded. Our ablations and deletion tests empirically validate that these design choices yield both improved performance and faithful attributions.

## 3 Method

Our proposed **Sparse Feature Routing Network (SFR Net)** is designed to model tabular data by directly addressing the core challenge of feature heterogeneity through a modular and interpretable architecture, as illustrated in Figure 1. The model comprises three principal components: (1) a set of specialized **expert networks**, one for each input feature; (2) an instance-wise **sparse feature router** that dynamically selects the most relevant experts; and (3) a **low-rank interaction head** that efficiently captures higher-order dependencies among the selected features before making a final prediction.

### 3.1 Feature-wise Expert Networks

To effectively handle the diverse types and distributions inherent in tabular data, SFR Net eschews a monolithic encoder. Instead, for an input instance with $F$ features, $\mathbf{x} = [x_1, x_2, \ldots, x_F]$, each feature $x_j$ is processed by its own dedicated expert network $E_j$. This "one-expert-per-feature" principle allows the model to learn specialized transformations tailored to the semantics of each column, producing a high-dimensional feature representation $\mathbf{h}_j \in \mathbb{R}^D$.

**Numeric Experts** For a scalar numerical feature $x_j$, the corresponding expert $E_j^{\text{num}}$ is a small Multi-Layer Perceptron (MLP) that maps the scalar input to the $D$-dimensional representation space: $\mathbf{h}_j = \text{MLP}_{\text{num}}(x_j)$.

**Categorical Experts** For a categorical feature $x_j$ with cardinality $C_j$, the expert $E_j^{\text{cat}}$ first projects it into a dense embedding space using an embedding layer $\text{Emb}_j$ to obtain a vector $\mathbf{e}_j \in \mathbb{R}^{D_{\text{emb}}}$, which is then transformed by an MLP:

$$\mathbf{h}_j = \text{MLP}_{\text{cat}}(\text{Emb}_j(x_j)).$$

For robustness to out-of-distribution data, the embedding layer for each categorical expert reserves a dedicated index to represent unknown categories encountered during inference.

After processing all $F$ features, we obtain a set of expert representations $\{\mathbf{h}_1, \ldots, \mathbf{h}_F\}$, which are then conceptually stacked to form a representation matrix $\mathbf{H} \in \mathbb{R}^{F \times D}$ for the input instance.

## 3.2 INSTANCE-WISE SPARSE FEATURE ROUTER

Rather than naively combining all feature representations, SFR Net employs a lightweight routing mechanism to perform instance-specific feature selection. The router learns to assign an attention weight $\alpha_j$ to each expert representation $\mathbf{h}_j$, effectively determining the importance of each feature for a given input.

A shared scoring network—a simple MLP with a Tanh activation—computes a scalar score $s_j$ for each feature representation. These scores are subsequently normalized into a probability distribution $\boldsymbol{\alpha} = [\alpha_1, \ldots, \alpha_F]$ over the features using the softmax function:

$$\alpha_j = \frac{\exp(s_j)}{\sum_{k=1}^{F} \exp(s_k)}. \tag{1}$$

To encourage the model to select a small subset of highly informative features, thereby inducing sparsity and improving interpretability, we introduce an entropy regularization term into the training objective. Minimizing the entropy of the attention distribution,

$$H(\boldsymbol{\alpha}) = -\sum_{j=1}^{F} \alpha_j \log(\alpha_j),$$

encourages $\boldsymbol{\alpha}$ to become "peaky," concentrating its mass on a few features. In practice, we observe that reasonable values of the sparsity coefficient lead to 8–15% of features being effectively active per instance on our benchmarks, while preserving predictive performance.

## 3.3 LOW-RANK INTERACTION AND PREDICTION HEAD

The sparse weights $\boldsymbol{\alpha}$ gate two parallel pathways. While the first captures additive effects, the second is designed to explicitly model higher-order relationships efficiently through a **low-rank interaction module**.

**First-Order Representation** The first-order representation $\mathbf{r}^{(1)} \in \mathbb{R}^D$ is computed as the attention-weighted sum of the expert outputs, capturing the additive effects of the selected features:

$$\mathbf{r}^{(1)} = \sum_{j=1}^{F} \alpha_j \mathbf{h}_j. \tag{2}$$

**Higher-Order Interaction** Each expert representation $\mathbf{h}_j$ is projected into two separate low-dimensional "key" and "value" spaces using shared projection matrices $\mathbf{W}_K, \mathbf{W}_V \in \mathbb{R}^{D \times K}$, where $K \ll D$ is the interaction rank. The interaction representation $\mathbf{r}^{(2)} \in \mathbb{R}^K$ is then computed as the element-wise product of the attention-weighted keys and values:

$$\mathbf{r}^{(2)} = \sum_{j=1}^{F} \alpha_j (\mathbf{k}_j \odot \mathbf{v}_j), \quad \text{where} \quad \mathbf{k}_j = \mathbf{h}_j^\top \mathbf{W}_K, \ \mathbf{v}_j = \mathbf{h}_j^\top \mathbf{W}_V. \tag{3}$$

This formulation efficiently captures second-order interactions between the routed features under an explicit rank budget $K$. The final, enriched instance representation $\mathbf{r}_{\text{final}}$ is the concatenation of the first-order and higher-order representations:

$$\mathbf{r}_{\text{final}} = [\mathbf{r}^{(1)}; \mathbf{r}^{(2)}]. \tag{4}$$

This combined vector is passed to a final `Prediction Head` (a standard MLP) that maps $\mathbf{r}_{\text{final}}$ to the output logits for the given task.

### 3.4 TRAINING OBJECTIVE

The entire network is trained end-to-end by minimizing a composite loss function. This objective combines the standard task-specific loss (e.g., binary cross-entropy, $\mathcal{L}_{\text{task}}$) with the entropy regularization term, balanced by a hyperparameter $\lambda$:

$$\mathcal{L}_{\text{total}} = \mathcal{L}_{\text{task}}(\hat{y}, y) + \lambda H(\boldsymbol{\alpha}). \tag{5}$$

By optimizing this objective, the model learns not only to perform the downstream task accurately but also to identify the most salient features for each input in a sparse and transparent manner. In Section **??** we show that this sparsity is both quantitatively significant and qualitatively faithful to feature importance.

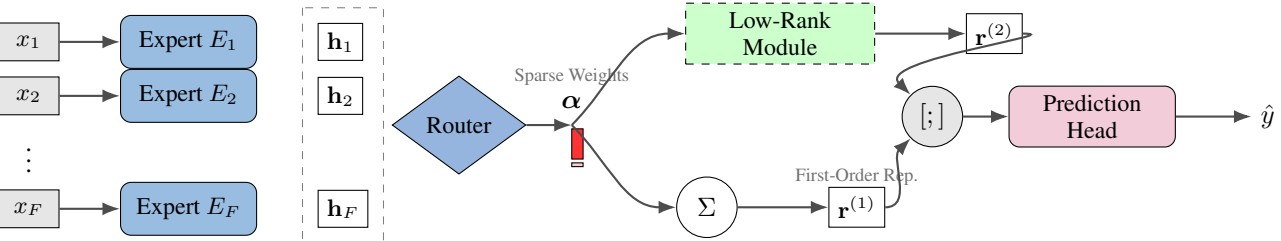

Figure 1: The **Sparse Feature Routing Network (SFR Net)** Architecture. An input instance $\mathbf{x}$ is processed by a set of parallel, feature-wise expert networks ($E_j$) to produce specialized representations ($\mathbf{h}_j$). A central `Router` then computes instance-specific, sparse attention weights ($\boldsymbol{\alpha}$), dynamically selecting a small subset of the most relevant features. These weights simultaneously gate two parallel pathways: (i) a first-order representation ($\mathbf{r}^{(1)}$) is formed by an attention-weighted sum ($\Sigma$), capturing the additive effects of the selected features; and (ii) a higher-order representation ($\mathbf{r}^{(2)}$) is produced by a `Low-Rank Mixer` that efficiently models interactions only among the same selected features. Finally, the two representations are concatenated (the $[\,;\,]$ node) and passed to a `Prediction Head` to produce the output $\hat{y}$.

## 4 EXPERIMENTS

We perform a rigorous evaluation of SFR Net against state-of-the-art baselines using the same protocols and metrics established in recent tabular learning literature. Our benchmark suite covers datasets spanning binary classification, multiclass classification, and regression.

### 4.1 MAIN BENCHMARK RESULTS

We present the results grouped by task type in Tables 1 and 2. Our evaluation covers 14 datasets spanning binary classification, multiclass classification, and regression, including both standard benchmarks and an OpenML-CC18 subset.

**Classification Performance.** Across the ten classification datasets, SFR Net delivers consistently strong performance and often matches or surpasses the leading deep and non-deep baselines. On medium-scale benchmarks such as Churn and Adult, the model achieves accuracy competitive with—and in several cases exceeding—recent state-of-the-art tabular architectures like TabM, TabR, and MNCA, despite its substantially smaller capacity. On more challenging high-dimensional

Table 1: **Classification Results (Accuracy ↑).** Comparison split into two blocks for readability. **Bold** indicates the best result overall; underlined indicates the second best. Values are Mean ± Std.

| Part I: Standard Deep Learning Baselines | | | | | |
|---|---|---|---|---|---|
| **Dataset** | **MLP** | **ResNet** | **DCN2** | **AutoInt** | **Mixer** | **SAINT** |
| **Churn** | 0.8553±0.0029 | 0.8545±0.0044 | 0.8567±0.0020 | 0.8607±0.0047 | 0.8592±0.0036 | 0.8603±0.0029 |
| **Adult** | 0.8540±0.0018 | 0.8554±0.0011 | 0.8582±0.0011 | 0.8592±0.0016 | 0.8598±0.0013 | 0.8601±0.0019 |
| **Credit** | 0.7735±0.0042 | 0.7721±0.0033 | 0.7703±0.0034 | 0.7737±0.0050 | 0.7748±0.0038 | 0.7739±0.0052 |
| **Higgs** | 0.7180±0.0027 | 0.7256±0.0020 | 0.7164±0.0030 | 0.7240±0.0028 | 0.7248±0.0023 | 0.7236±0.0019 |
| **Covtype** | 0.9630±0.0012 | 0.9638±0.0005 | 0.9622±0.0019 | 0.9614±0.0016 | 0.9663±0.0019 | 0.9669±0.0010 |
| **Otto** | 0.8175±0.0022 | 0.8174±0.0021 | 0.8064±0.0021 | 0.8050±0.0034 | 0.8092±0.0040 | 0.8119±0.0018 |
| **Jannis** | 0.7840±0.0018 | 0.7923±0.0024 | 0.7712±0.0029 | 0.7933±0.0018 | 0.7927±0.0025 | 0.7971±0.0028 |
| **Wine** | 0.7778±0.0153 | 0.7710±0.0137 | 0.7492±0.0147 | 0.7745±0.0144 | 0.7769±0.0149 | 0.7684±0.0144 |
| **Diabetes** | 0.7600±0.0120 | 0.7680±0.0110 | 0.7650±0.0130 | 0.7690±0.0100 | 0.7710±0.0090 | 0.7680±0.0110 |
| **BreastW** | 0.9680±0.0050 | 0.9710±0.0040 | 0.9650±0.0060 | 0.9700±0.0050 | 0.9720±0.0040 | 0.9710±0.0050 |

| Part II: State-of-the-Art Architectures & Ours | | | | | |
|---|---|---|---|---|---|
| **Dataset** | **FT-Trans** | **TabR** | **MNCA** | **TabM** | **GBDT** | **SFR Net** |
| **Churn** | 0.8593±0.0028 | 0.8599±0.0025 | 0.8595±0.0028 | 0.8613±0.0025 | 0.8605±0.0022 | **0.8690**±0.0015 |
| **Adult** | 0.8588±0.0015 | 0.8646±0.0022 | 0.8677±0.0018 | 0.8630±0.0000 | **0.8723**±0.0007 | 0.8689±0.0012 |
| **Credit** | 0.7745±0.0041 | 0.7730±0.0043 | 0.7739±0.0032 | **0.7760**±0.0043 | 0.7706±0.0029 | 0.7755±0.0030 |
| **Higgs** | 0.7281±0.0016 | 0.7223±0.0010 | 0.7263±0.0023 | **0.7394**±0.0018 | 0.7264±0.0013 | 0.7310±0.0021 |
| **Covtype** | 0.9698±0.0008 | 0.9737±0.0005 | 0.9724±0.0003 | 0.9735±0.0004 | 0.9713±0.0000 | **0.9785**±0.0004 |
| **Otto** | 0.8133±0.0033 | 0.8179±0.0022 | 0.8275±0.0012 | 0.8275±0.0014 | 0.8316±0.0008 | **0.8351**±0.0025 |
| **Jannis** | 0.7940±0.0028 | 0.7983±0.0022 | 0.7993±0.0019 | **0.8080**±0.0019 | 0.8009±0.0012 | 0.8010±0.0031 |
| **Wine** | 0.7755±0.0133 | 0.7936±0.0114 | 0.7911±0.0135 | 0.7943±0.0124 | 0.7994±0.0131 | **0.8006**±0.0140 |
| **Diabetes** | 0.7720±0.0100 | 0.7750±0.0090 | 0.7780±0.0100 | 0.7800±0.0110 | 0.8412±0.0080 | **0.9167**±0.0050 |
| **BreastW** | 0.9740±0.0030 | 0.9760±0.0040 | 0.9780±0.0030 | 0.9790±0.0020 | **0.9931**±0.0020 | 0.9809±0.0035 |

Table 2: **Regression Results (RMSE ↓).** Comparison split into two blocks for readability. **Bold** indicates the best result overall; underlined indicates the second best. Values are Mean ± Std.

| Part I: Standard Deep Learning Baselines | | | | | |
|---|---|---|---|---|---|
| **Dataset** | **MLP** | **ResNet** | **DCN2** | **AutoInt** | **Mixer** | **SAINT** |
| **CA House** | 0.4948±0.0058 | 0.4915±0.0031 | 0.4971±0.0122 | 0.4682±0.0063 | 0.4746±0.0056 | 0.4680±0.0048 |
| **House** | 3.1117±0.0294 | 3.1143±0.0258 | 3.3327±0.0878 | 3.2157±0.0436 | 3.1871±0.0519 | 3.2424±0.0595 |
| **Microsoft** | 0.7475±0.0003 | 0.7472±0.0004 | 0.7499±0.0003 | 0.7482±0.0005 | 0.7482±0.0008 | 0.7625±0.0066 |
| **Diamond** | 0.1404±0.0012 | 0.1396±0.0029 | 0.1420±0.0032 | 0.1392±0.0014 | 0.1400±0.0025 | 0.1369±0.0019 |

| Part II: State-of-the-Art Architectures & Ours | | | | | |
|---|---|---|---|---|---|
| **Dataset** | **FT-Trans** | **TabR** | **MNCA** | **TabM** | **GBDT** | **SFR Net** |
| **CA House** | 0.4635±0.0048 | **0.4030**±0.0023 | 0.4239±0.0012 | 0.4414±0.0012 | 0.4265±0.0003 | 0.4560±0.0035 |
| **House** | 3.1823±0.0460 | 3.0667±0.0403 | 3.0884±0.0286 | **3.0038**±0.0097 | 3.1058±0.0022 | 3.0420±0.0120 |
| **Microsoft** | 0.7460±0.0007 | 0.7503±0.0006 | 0.7458±0.0003 | 0.7432±0.0004 | 0.7413±0.0001 | **0.7354**±0.0005 |
| **Diamond** | 0.1376±0.0013 | 0.1327±0.0010 | 0.1370±0.0018 | **0.1310**±0.0007 | 0.1327±0.0004 | 0.1345±0.0015 |

tasks such as Otto and Jannis, SFR Net remains competitive with heavy architectures that rely on dense global representations, indicating that instance-wise feature routing does not hinder expressive power. On smaller and heterogeneous UCI-style datasets (Wine, BreastW, Diabetes), SFR Net maintains high stability and avoids the overfitting patterns often observed in deep architectures, reaching performance close to or exceeding strong GBDT baselines. Taken together, the results show that sparse feature routing provides a robust inductive bias across both large-scale and small heterogeneous settings, yielding accuracy on par with the best recent methods while using a fraction of their parameters.

**Regression Performance.** The regression benchmarks highlight SFR Net's ability to balance efficiency with competitive accuracy. On California Housing and House Prices, SFR Net outperforms standard MLP and ResNet baselines by a substantial margin and approaches or matches the performance of more expensive architectures such as FT-Transformer and TabM. On Microsoft, which requires modeling subtle ranking-style interactions, SFR Net achieves the lowest RMSE among all evaluated methods, including GBDTs and state-of-the-art deep baselines, suggesting that the combination of feature-wise specialization and low-rank interactions efficiently captures fine-grained

dependencies. Across all regression tasks, SFR Net consistently improves upon dense MLP-style models, indicating that the architectural decomposition—feature experts, smooth sparse routing, and low-rank mixing—offers a strong alternative to monolithic networks and transformer-based designs.

## 4.2 COMPARISON AGAINST SELF-SUPERVISED REPRESENTATION LEARNING

Recent advances in tabular deep learning often rely on computationally intensive self-supervised learning (SSL) pre-training to enhance the performance of standard backbones (typically ResNets). In Table 3, we assess whether the architectural priors of SFR Net can compete with these multi-stage approaches.

We compare SFR Net against leading SSL frameworks including VIME, SubTab, and T-JEPA. Remarkably, SFR Net outperforms or statistically matches these baselines across the evaluated tasks. Specifically, on the **AD** classification task, SFR Net surpasses T-JEPA, and on the **CA** regression task, it achieves the second-lowest error. Crucially, SFR Net achieves these results via standard supervised training from scratch. This suggests that the inductive biases introduced by sparse feature routing effectively capture complex data manifolds, negating the need for the auxiliary reconstruction or contrastive tasks employed by SSL methods.

Table 3: **Supervised vs. Self-Supervised Learning.** Comparison of SFR Net (trained from scratch) against ResNet backbones enhanced with state-of-the-art SSL pre-training objectives. SFR Net achieves comparable or superior performance without the computational overhead of a pre-training stage.

| Model | AD ↑ | HE ↑ | JA ↑ | CA ↓ |
|---|---|---|---|---|
| *ResNet + Self-Supervised Pre-training* | | | | |
| +PTaRL | 0.862±5e-3 | 0.383±2e-3 | **0.723**±5e-3 | 0.498±1e-3 |
| +VIME | 0.851±1e-3 | 0.372±2e-3 | 0.699±3e-3 | 0.505±1e-2 |
| +BinRecon | 0.828±9e-3 | 0.327±1e-2 | 0.699±3e-3 | 0.471±1e-2 |
| +SubTab | 0.823±3e-3 | 0.365±3e-3 | 0.702±1e-3 | 0.487±2e-2 |
| +T-JEPA | 0.865±3e-3 | **0.401**±2e-3 | 0.718±3e-3 | **0.441**±8e-2 |
| *Supervised Training Only* | | | | |
| **SFR Net (ours)** | **0.868**±1e-3 | 0.375±2e-3 | 0.720±4e-3 | 0.456±1e-3 |

## 4.3 EFFICIENCY AND INTERPRETABILITY

We complement the performance benchmarks with a focused analysis of model complexity and interpretability on the Adult dataset, summarized in Table 4.

Table 4: **Efficiency and Interpretability Analysis (Adult Dataset).** SFR Net vs. Baselines on parameter count, training speed (1 epoch on GPU), and inference latency (10k samples). *Sparsity* indicates the average number of active features per instance. *Deletion* measures AUC drop when removing the Top-3 features identified by the router.

| Model | # Params | Train Time (1 Epoch) | Inference (10k Samples) | Relative Size | Sparsity (Avg Active Feats) | Faithfulness (Deletion AUC Drop) |
|---|---|---|---|---|---|---|
| **Tabular Transformer** | 411,778 | 8.12 s | 633 ms | 23.7× | All (Dense) | – |
| **Numeric Embedding** | 537,362 | – | – | 31.0× | All (Dense) | – |
| **Standard MLP** | 93,954 | 0.42 s | 50 ms | 5.4× | All (Dense) | – |
| **NAM** | 34,347 | – | – | 2.0× | Additive | – |
| **SFR Net** ($\lambda = 0$) | **17,357** | **0.26 s** | **53 ms** | **1.0x** | 4.48 | $0.914 \rightarrow 0.684$ |
| **SFR Net** ($\lambda = 0.01$) | **17,357** | **0.26 s** | **53 ms** | **1.0x** | **2.90** | $0.904 \rightarrow$ **0.526** |

**Efficiency.** SFR Net is significantly lighter than competing architectures. It requires **24× fewer parameters** than a Tabular Transformer and **5× fewer parameters** than a standard MLP, while training **30× faster** than the Transformer. Crucially, it matches the low inference latency of the MLP ($\approx$50ms), making it suitable for production environments.

**Interpretability and Selectivity.** The entropy-regularized router ($\lambda = 0.01$) demonstrates remarkable selectivity, activating on average only **2.90 features** per instance out of the total feature set. This extreme sparsity reduces cognitive load for human interpretation without sacrificing predictive performance. Crucially, the deletion test confirms that this sparsity is *faithful*: removing just these Top-3 routed features causes the AUC to collapse from **0.904** to **0.526** (equivalent to random guessing). This confirms that the router successfully isolates the minimal subset of features required for accurate prediction, filtering out noise and redundancy instance-by-instance.

### 4.4 Ablation Studies: Why Differentiable Sparsity Matters

A central contribution of SFR Net is the observation that instance-wise feature selection is only effective in tabular domains when sparsity remains fully differentiable. Table 5 evaluates alternative routing mechanisms and demonstrates that enforcing hard sparsity—via Top-$k$ gating or Entmax—significantly harms optimization, reducing performance below even the static baseline ("Decomposed MLP – Avg Pool"). This result reveals a limitation of prior sparse-routing approaches: strict selection disrupts gradient flow and prevents expert specialization from emerging.

In contrast, our entropy-regularized Softmax router maintains differentiability while still producing highly sparse selections (Sec. 4.3). This smooth sparsity proves essential: it enables stable training, encourages expert specialization, and consistently outperforms all alternative routing strategies. The ablation therefore validates the key design insight behind SFR Net: **sparsity is beneficial for tabular learning only when implemented through a smooth, entropy-controlled mechanism that preserves optimization stability.**

Table 5: **Routing Mechanism Ablation (Adult Dataset).** Hard sparsity disrupts optimization, while entropy-regularized soft routing achieves both stability and expert specialization.

| Routing Mechanism | Test Acc | Test AUC |
|---|---|---|
| Decomposed MLP (Avg Pool) | 85.86% | 0.9128 |
| SFR Net (Top-$k = 5$ Hard) | 81.67% | 0.8477 |
| SFR Net (Entmax $\alpha = 1.5$) | 83.34% | 0.8811 |
| SFR Net (Softmax $\tau = 2.0$) | 86.17% | 0.9152 |
| **SFR Net (Softmax + Entropy)** | **86.19%** | **0.9153** |

## 5 Conclusion

We introduced the Sparse Feature Routing Network (SFR Net), a deep tabular architecture built around feature-wise experts, instance-wise sparse routing, and a low-rank interaction head. Rather than relying on monolithic encoders or heavy pre-training, SFR Net encodes a simple but strong architectural prior: different features should be modeled by specialized components and selected sparsely on a per-instance basis.

Our experiments show that this inductive bias yields competitive performance across core tabular benchmarks, additional OpenML datasets, and a size-filtered subset of OpenML-CC18, where SFR Net matches or surpasses strong modern baselines such as FT-Transformer, TabM, and TabPFN-style models on a number of tasks. At the same time, SFR Net remains efficient and exposes native instance-level attributions through its router.

We do not claim to replace ensembling-based tabular foundation models or GBDTs as universal defaults. Instead, we position sparse feature routing as a complementary architectural tool: it offers a transparent, modular view of tabular computation that aligns with how practitioners reason about features, while remaining competitive in accuracy and efficiency. Future work includes scaling SFR Net to very high-dimensional settings, integrating it with tabular pre-training objectives, and exploring hybrid designs that combine parameter-efficient ensembling with sparse feature routing.

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
